# Patients with Chronic Lymphocytic Leukemia Have a Very High Risk of Ineffective Response to the BNT162b2 Vaccine

**DOI:** 10.3390/vaccines10071162

**Published:** 2022-07-21

**Authors:** Andrea Galitzia, Luca Barabino, Roberta Murru, Giovanni Caocci, Marianna Greco, Giancarlo Angioni, Olga Mulas, Sara Oppi, Stefania Massidda, Alessandro Costa, Giorgio La Nasa

**Affiliations:** 1Department of Medical Sciences and Public Health, University of Cagliari, 09042 Monserrato, Italy; luca.barabino@hotmail.it (L.B.); olga.mulas@aob.it (O.M.); alexcosta16@hotmail.it (A.C.); giorgio.lanasa@aob.it (G.L.N.); 2Hematology and Transplant Centre, Ospedale Oncologico Armando Businco, ARNAS G. Brotzu, 09121 Cagliari, Italy; roberta.murru@aob.it (R.M.); marianna.greco@aob.it (M.G.); sara.oppi@aob.it (S.O.); stefania.massidda@aob.it (S.M.); 3Laboratory of Clinical Chemical Analysis and Microbiology, ARNAS G. Brotzu, 09134 Cagliari, Italy; giancarlo.angioni@aob.it

**Keywords:** COVID-19, chronic lymphocytic leukemia (CLL), vaccine, hypogammaglobulinemia, immunity, immunocompromised

## Abstract

Patients with CLL have high rates of either severe disease or death from COVID-19 and a low response rate after COVID-19 vaccination has been reported. We conducted a single-center study with the main objective to evaluate the immunogenicity of the BNT1162b2 mRNA vaccines in 42 patients affected by CLL with the assessment of antibody response after the second and the third dose. After the second dose of vaccine, 13 patients (30%) showed an antibody response. The presence of hypogammaglobulinemia and the use of steroids or IVIG were the main factors associated with poor response. After the third dose, 5/27 (18%) patients showed an antibody response while in non-responders to the second dose, only 1 patient (4%) showed an elicitation of the immune response by the third dose, with no statistically significant difference. Our data, despite the small size of our cohort, demonstrate that patients with CLL have a low rate of effective response to the BNT162b2 vaccine. However, the effective role of a subsequent dose is still unclear, highlighting the need for alternative methods of immunization in this particularly fragile group of patients.

## 1. Introduction

Among hematological diseases, chronic lymphocytic leukemia (CLL) is characterized by deep immunological dysregulation which leads to profound immunosuppression due to both therapy-related factors and patient characteristics as well as factors intrinsic to the disease [1,2]. Patients with CLL are reported to have high rates of both severe disease and death for COVID-19 [3,4]. In addition, poor serological response to the SARS-CoV-2 vaccines in those patients has already been described [2,5]. However, immunocompromised patients were excluded from early trials of COVID-19 immunization. Subsequent prospective trials have shown a low response rate, ranging from nearly 50% of treatment-naïve patients to 16% in those undergoing active treatment after two mRNA vaccine doses [6,7]. A third BNT162b2 mRNA COVID-19 vaccine dose could be effective in 25–35% of patients who failed to respond to two doses [8,9,10,11]. However, the impact of repeated doses on antibody response is still unclear in these patients and, in particular, in those undergoing active treatment. This study aimed to define the humoral response in 42 CLL patients after two mRNA vaccine doses and subsequently the role of a third dose.

## 2. Materials and Methods

We carried out a single-center study to define the humoral response in 42 CLL patients after two BNT162b2 mRNA vaccine doses (30 µg per dose), administered according to the label, between March and June 2021 and, subsequently, after a third dose administered between September 2021 and January 2022. Patients with anamnestic SARS-CoV-2 detection prior to the first dose were excluded. Written informed consent was collected according to the Declaration of Helsinki.

Neutralizing antibodies titer detection was achieved by @Liaison^®^ TrimericS IgG Diasorin (sensitivity 98.7%, specificity 99.5%), at least 30 days after the second and the third dose, following the manufacturer’s instructions [12]. Assays in Arbitrary Units/mL (AU/mL) were converted into international standard units (BAU/mL) using a conversion factor of 2.6 according to the manufacturer’s recommendations. Patients with BAU/mL < 33.8 were considered non-responders while those with titer > 33.8 BAU/mL were defined as responders. 

We also performed immunophenotyping of peripheral blood before both the first dose and the third dose, in order to establish possible correlations between T, B, and NK lymphocyte subsets and single patient immune response. BD Multitest™ 6-Color TBNK with o BD Trucount™ Tubes was used with BD FACSLyric™ flow cytometers to determine the percentages and absolute counts of the following mature human lymphocyte subsets in peripheral whole blood for immunophenotyping: T-lymphocytes (CD3+), B-lymphocytes (CD19+), natural killer (NK) lymphocytes (CD3– CD16+ and/or CD56+), helper/inducer T-lymphocytes (CD3+CD4+), and suppressor/cytotoxic T-lymphocytes (CD3+CD8+). TBNK contains FITC-labeled CD3, clone SK7; PE-labeled CD16, clone B73.1 and PE-labeled CD56, clone NCAM16.2; PerCP-Cy 5.5-labeled CD45, clone 2D1 (HLe-1); PE-Cy7-labeled CD4, clone SK3; APC-labeled CD19, clone SJ25C1;25 and APC-Cy7-labeled CD8, clone SK1. The gating strategy is that shown in Figure 1.

A univariate analysis was performed to correlate antibody response and clinical and biological factors, in particular, Binet stage of disease at vaccination time, Cumulative Illness Rating Scale (CIRS) score, history of previous infections in the 12 months before the administration of the first dose of vaccine, CD19/CD4/CD8/CD3/NK counts, FISH Dohner’s category, TP53 status, IGHV mutational status, NOTCH1 mutation, number of previous lines of therapy, hypogammaglobulinemia at time of vaccination, steroid therapy, intravenous immunoglobulin (IVIG). Categorical variables were compared using the chi-square test and Fisher’s exact test with a binary logistic regression model. Continuous variables were compared with the Mann–Whitney test. *p*-values < 0.05 were considered statistically significant. All the analyses were performed using the statistical software Prism 5.04. The status of all included patients was updated on 15 May 2022. Patient characteristics and results are summarized in Table 1.

## 3. Results

The median age at the time of vaccination was 68 years (range 40–84), with CLL diagnosed between 1997 and 2021. According to Binet stage 5 patients (12%) were stage A, 28 (67%) B, and 9 (21%) C. The median CIRS score was 3 (range 0–15).

Regarding any previous infections, 19 patients (44.5%) had at least one event (all grades of severity according to CTCAE scale) within 12 months before the first dose of vaccination.

Only 5 patients (12%) were treatment-naïve; 37 patients (88%) received at least one line of therapy and among them 78% (n° 29) were on active treatment. Treatment included chemoimmunotherapy (n° 5; 12%), BTK inhibitor (n° 21; 50%), venetoclax plus rituximab (n° 7; 17%) and steroids (n° 4; 9%). 

After the second dose of vaccine, 13 patients (31%) showed adequate levels of anti-SARS-CoV-2 IgG, with a median titer of 382.38 BAU/mL (51.22–1040); otherwise, 29 (69%) were seronegative (Figure 2). 

Factors associated with a poor humoral response were the presence of hypogammaglobulinemia in at least one immunoglobulin subtype (IgG < 700 mg/dL; IgM < 40 mg/dL; IgA < 70 mg/dL) [OR 0.22 (95%CI 0.049–0.99) *p* = 0.045], and prolonged steroid therapy or need for IVIG [OR 0.16 (95%CI 0.031–0.88) *p* = 0.018].

Moreover, there was no correlation between IgG and IgA levels and anti SARS-CoV-2 neutralizing antibodies [Spearman r 0.01(95%CI −0.31 to 0.33) *p*= 0.94 for IgG, Spearman r 0.2 (95%CI −0.12 to 0.49) *p*= 0.21 for IgA], whereas a correlation was found for IgM levels [Spearman r 0,32 (95%CI 0.008 to 0.59) *p*= 0.04].

Levels of antibodies were significantly lower in patients with hypogammaglobulinemia (at least one class of Ig), with a mean of 99.1 BAU/mL [95%CI 6.52–191.8] vs. 214.6 BAU/mL [95%CI 4.81–457.8 *p* = 0.022] in patients with IgM levels < 40 mg/dL [41.91 BAU/mL (95%CI 7.48–76.34) vs. 239.6 BAU/mL (95%CI 43.88–435.3) *p* = 0.029] in those with an NK count <300/µL [66.39 BAU/mL (95%CI 4.81–145.4) vs. 233.6 BAU/mL (95%CI 33.47–413.7), *p* = 0.047], and in those receiving steroids or IVIG [34.09 BAU/mL (95%CI 4.81–77.85) vs. 184.2 BAU/mL (95%CI 47.92–320.5) *p* = 0.005)]. On the other hand, we did not find any statistical correlation between IgG SARS-CoV-2 titers, the type of therapy (i.e.,: BTKi, Ven-R), and timing from the last rituximab. However, it should be observed that there was an undoubted unfavorable effect for Ven-R and time < 12 months from last MoAb anti CD20 (Figure 3).

Even though not statistically significant, flow cytometry for the TBNK profile showed some difference between responders and non-responders after the second vaccine dose. There was a substantial overlap of expression for CD3/CD8, while a different pattern for CD19 intensity has been observed, likely attributable to the expansion of clonal ineffective CD19+ lymphocyte in non-responders with active disease and on treatment (Figure 1).

CIRS score (*p* = 0.61), type of therapy (*p* = 0.45), number of previous treatments (*p* = 0.18), absolute lymphocyte count (*p* = 0.89), and CD19/CD3/CD4 subsets (*p* = 0.45; *p* = 0.86; *p* = 0.68, respectively) were not statistically significant.

Of 42 patients, 39 received the third dose, with 27 evaluable for analysis; of those, no one was treatment-naïve, 5 were off-therapy, and 22 were on active treatment with BTKi or venetoclax and rituximab. Furthermore, 5 were already responders to the second dose. Of 12 patients not analyzed, 7 had COVID-19 and 1 refused; the others had lost contact. The response rate after the second and the third dose was not different. For non-responder to the second dose, only one patient (4%) responded to the third, while one defined as a responder became negative after the third one (Figure 2).

Factors associated with a poor response after the third dose were the presence of anemia (*p* = 0.031), a history of infection before the vaccine (*p* = 0.014), and the last administration of anti-CD20 MoAb less than 12 months before the vaccination (*p* = 0.044).

Mild adverse events after vaccination were reported in seven patients (17%), mainly injection site pain. 

We reported seven COVID-19 infections after vaccination: one in a responder to the second dose (asymptomatic infection) and six in non-responder patients, with mild to severe disease that required hospital admission in four of them, with need for respiratory support. Of those, two patients died of COVID-19 pneumonia. 

Among our cohort, two other deaths have been reported during follow-up, due to other causes than COVID-19 (one due to Richter’s syndrome, one due to metastatic lung cancer).

## 4. Discussion

Randomized controlled trials demonstrated that the BNT162b2 vaccine is highly effective in a healthy population, with a response rate of 91% after two doses [13]. Patients with hematological malignancies had different seroconversion rates, ranging from 94% in myelodysplastic syndrome and Hodgkin lymphoma to 47% in CLL [14]. 

Our data, even though based on a small patient cohort, highlight that CLL patients have a very low rate of response to SARS-CoV-2 immunization and a very high risk for severe disease and death due to COVID-19, according to what was previously reported in literature so far [5,6,15,16,17,18]. In particular, patients on active treatment with target therapies and anti-CD20 MoAb are at high risk for low rate of response [8,9,19]. After the second dose, untreated patients had a response rate of 60% versus 23% in treated ones, underlining the previously reported difference in seroconversion [7,14]. Our data globally agree with findings of Roeker LE et al. in a similar cohort that reported a seroconversion rate of 23% in treated patients (21% BTKi; 14% anti-CD20 MoAb; 0% Ven-R) [19].

Immunocompromised status related to CLL disease, frequently associated with hypogammaglobulinemia and, in addition, related to therapies, is the main cause of the poor response to vaccination [1,17].

Although in our analysis we did not find any significant statistical correlation between disease stage, number of previous therapies, and humoral immune response (bias due to small sample size), we can assume that both conditions adversely impact the immunological response to the vaccine, as reported in literature [18,20].

With regard to the repeated doses after the second one, we did not find any benefit for the third dose in our cohort (Figure 1), even though data are limited by the small size of the cohort. These findings agree with what was reported by Kohn M. et al. in 33 CLL patients treated with anti-CD20 MoAb [21]. Moreover, Bagacean C. et al. reported a global response rate of 35% after administration of third dose to patients that failed the second, with a lower rate (24%) for patients on active treatment [9]. Herishanu Y. et al. reported a rate of 24% that was lower in patients subject to treatment with BTKi (15%) and Ven-R (7%) [8]. In our cohort, nearly 80% of patients who failed the third dose were on active treatment with BTKi or BCL2i plus rituximab, suggesting that those patients are at the highest risk to fail immunization against SARS-CoV-2.

Taken together, our data suggest that, currently, immunization against SARS-CoV-2 is still an unmet medical need for patients with CLL and, in particular, in those on active treatment or with strong immunodepression associated with extensive hypogammaglobulinemia. We believe that optimizing humoral response with specific vaccination strategies should be subject to current and future investigation, as well as subsequent boost vaccination, passive immunization with monoclonal antibodies against SARS-CoV-2 (i.e., tixagevimab/cilgavimab [22]), and, eventually, pausing of concomitant immunosuppression.

## Figures and Tables

**Figure 1 vaccines-10-01162-f001:**
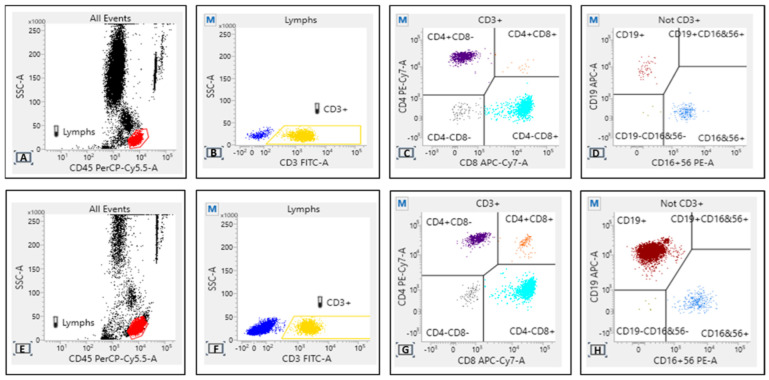
Flow cytometry density plot for responders (**A**–**D**) and non-responders (**E**–**H**). Some differences should be assumed for CD19 expression, likely attributable to the expansion of clonal lymphocytes in non-responders with active disease and on treatment.

**Figure 2 vaccines-10-01162-f002:**
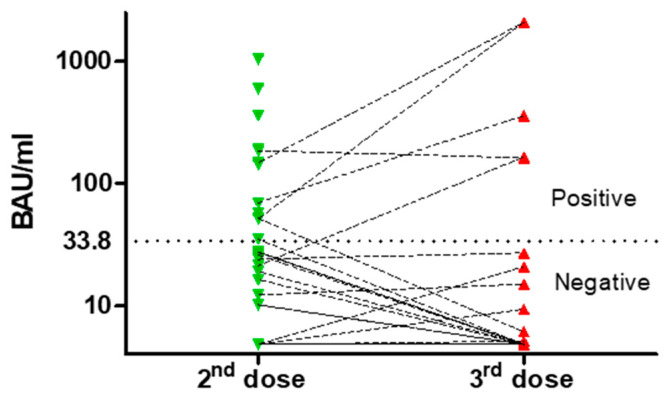
Neutralizing antibody titers after second and third dose. The dotted line represents the cut-off value between responders and non-responders.

**Figure 3 vaccines-10-01162-f003:**
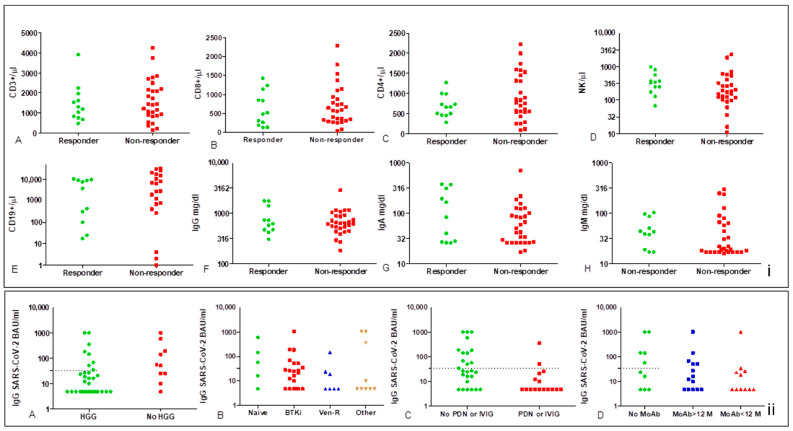
Dot-plots of lymphocyte subset count and Ig levels according to the status of responder and non-responder (**i**) and IgG-SARS-CoV-2 titers (**ii**). There is no statistically significant difference in lymphocyte subsets between responders and non-responders (**i**); a gap of distribution in the normal range could be observed in CD19 count for non-responders (**i.E**). Neutralizing antibody titers were significantly low in patients with hypogammaglobulinemia (**ii.A**) and in those receiving steroids or IVIG (**ii.B**,**C**), Mann–Whitney *p* = 0.022 and *p* = 0.005, respectively, no significant correlation was found with the type of therapy and time from last rituximab, even though Ab titers and rate of response were very low in patients on Ven-R and in those treated with rituximab < 12 months (**ii.B**,**D**).

**Table 1 vaccines-10-01162-t001:** Baseline characteristics of patients and summary of response-associated factors.

	After 2nd Dose (n = 42)			After 3rd Dose (n = 27)		
Response to COVID-19 Vaccine	YES (n = 13)	NO (n = 29)	*p*	YES (n = 5)	NO (n = 22)	*p*
Sex						
M	4	13	0.39	5	10	**0.03**
F	9	16		0	12	
Previous Infections						
Yes	5	13	0.69	0	11	**0.04**
No	8	16		5	11	
Age			0.69			0.55
>65	5	15		2	12	
<65	8	14		3	10	
CIRS *			0.61			0.97
<3	7	18		3	13	
>3	6	11		2	9	
Stage Binet			0.28			0.99
A	3	2		0	0	
B	9	22		4	18	
C	1	5		1	4	
N° of treatments			0.18			0.99
0–1	9	12		2	8	
>1	4	17		3	14	
Type of treatment			0.41			0.45
Naïve	3	2		-	-	
Other	3	6		0	5	
BTKi	6	15		4	12	
Ven-R	1	6		1	5	
Last rituximab			0.18			0.12
No rituximab	7	10		2	3	
>12months	5	9		3	9	
<12months	1	10		0	10	
PDN or IVIG *			**0.04**			0.29
No	11	15		4	12	
Yes	2	14		1	10	
ALC *			0.89			0.38
>5000/mm^3^	6	14		1	9	
<5000/mm^3^	7	15		4	13	
CD3			0.86			0.23
>690/mm^3^	10	23		0	5	
<690/mm^3^	3	6		5	17	
CD4			0.68			0.30
>410/mm^3^	11	23		0	4	
<400/mm^3^	2	6		5	18	
CD19			0.45			0.97
>20/mm^3^	10	19		3	13	
<20/mm^3^	3	10		2	9	
NK			0.10			0.73
>300/mm^3^	7	8		1	16	
<300/mm^3^	6	21		4	6	
HGG *			**0.02**			
No	6	4		1	3	0.71
Yes	7	25		4	19	

* CIRS: Cumulative Illness Rating Scale, PDN: prednisone, BTKi: Bruton kinase inhibitors, Ven-R: venetoclax plus rituximab, ALC: absolute lymphocytes count, HGG: hypogammaglobulinemia; *p*-values < 0.05 are in bold.

## Data Availability

Data supporting the reported results can be obtained by contacting the corresponding author.

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
