# Peer review of "Patients with Chronic Lymphocytic Leukemia Have a Very High Risk of Ineffective Response to the BNT162b2 Vaccine"

_vaccines, 2022, doi:10.3390/vaccines10071162_

Round 1

Reviewer 1 Report

This is a well-written manuscript, with a clear objective and an appropriate design. I believe that the paper has sufficient quality for publication, and the data is relevant for physicians. CLL patients receive treatments that interfere with immune responses. Therefore, a study evaluating serological responses to vaccination in CLL patients is of high interest. The results look good, and the statistics are appropriate. The strength of the paper is the specific subject. It is critical to know the degree of protection of cancer patients subjected to Covid19 vaccination. The weakness, the authors themselves pointed it in the manuscript. The relatively low number of patients. But we have to consider that CLL is not a very frequent cancer type in the population.

The only thing that the authors must improve is the presentation. To provide the data in a single table together with the statistics decreases the paper´s appeal. I would advise the authors to:

1. Show direct flow cytometry data for their immune cell quantifications, together with dot plots for the individual patients. The authors provide all the data regarding quantifications of CD3, CD4, CD19 and NK cells. For these quantifications, the authors must present flow cytometry histograms or density plots on these data, at least one representative example for each population in patients that responded well to vaccination and those who did not. In these plots, I would suggest them to show the gating strategy, and the expression patterns of the specific markers used to identify these cells.

2. Represent the data in graphs, which should improve the quality of the presentation. Additionally, I would advise the authors to provide the specific quantitative data shown in the table for each patient in the form of a dotplot. For example, the authors show the number of patients that are above or below of a pre-stablished cutoff value. They do that to assess statistical differences. However, the reader would surely like to see the actual quantitative data for each patient in the form of a dotplot. This will also provide information on the variability found in each group.

Author Response

We would like to thank you for your kind review that allowed us to improve our manuscript in different ways. Here we provide detailed answers to your comments:

  • We agree with your observations regarding data representation, so we changed it by making graphics as you asked. We give a representation of the flow cytometric TBNK profile for one representative patient for responders and non-responders. [Figure 2]
  • We provide graphics about levels of immune cells and immunoglobulins in responders and non-responders and also titers of neutralizing antibodies according to the presence of each type of treatment and time from rituximab therapy. [Figure 3]

Reviewer 2 Report

This manuscript highlights an important risk for CLL patients regarding their response to the SARS-CoV-2 vaccine BNT162b2. However, I have major concerns regarding the presentation of the data and relevance to what is already known. Unfortunately, I do not think this manuscript is suitable for publication in this form.

Minor comments:

1.    In M&M lines 47-48, it would have been better to stress that you measure neutralizing antibodies with the Liaison TrimericS IgG Diasorin assay.

2.    In M&M line 58, you state history of previous infections in the last 12 months. Is this counted before the first dose or before the second dose?

3.    English writing should be checked and improved.

Major comments:

1.     Table 1 seems to have many mistakes, since often the numbers for several characteristics do not add up to the total number in that group. For example, for Stage Binet there are 14 patients (3+9+2) listed as responders after 2nd dose whereas total number in that group is 13. Or 36 patients (8+24+4) listed as non-responders after 2nd dose in a group that should be a total of 29. And there are other times where the numbers do not add up in the table. Also, why are some numbers listed as “0” and others as “-“? Also, should the p value 0.03 (sex, 3rd dose) be in bold? Are there patients not on rituximab?

2.     There is no data representation of the measured neutralizing IgG titers, although it is described in Result section in a very summarized way (lines 81-83). There should have been graphs showing the AU/mL for all patients after 2nd dose and after 3rd dose.

3.     Data is described in Results section for the measurements of IgG, IgM and IgA, but is too much summarized in data table under HGG (hypogammaglobulinemia). Are there any correlations with those titers with titers of neutralizing SARS-CoV-2 besides the measured significance for patients after the 2nd dose?

4.     In Results lines 96-99, it is described that of the 42 patients after the 2nd dose, 39 received a third dose of which 27 were analyzed. Thus 12 of these 39 were not analyzed. In the text it is written 13 patients were not analyzed?

5.     Are the 27 analyzed patients after the 3rd dose part of the 29 non-responders after the 2nd dose or is it a mix of responders and non-responders from the 2nd dose? Are there any responders after the 2nd dose that became non-responder?

6.     A weakness of this cohort is that there is no control cohort. There is no mention of the seroconversion percentage in “healthy” individuals.

7.     A weakness there are several publications already published regarding seroconversion in CLL patients after vaccination for SARS-CoV-2, for example: Herishanu et al. ~40% seroconversion (doi: 10.1182/blood.2021011568), Bergman et al. ~63% (doi: 10.1016/j.ebiom.2021.103705), Herzog Tzarfat et al. ~47% (10.1002/ajh.26284), Roeker et al. ~52% (doi: 10.1038/s41375-021-01270-w). How does the data of this manuscript relate to already published data? Are there differences/similarities with these studies. What makes your cohort unique? Did you look to different patient characteristics parameters? This needs to be discussed!

Author Response

We would like to thank you for your meticulous review that allowed us to improve our manuscript in different ways. Here we provide detailed answers to your comments.

Major comments:

  1. We found a few typos in Table 1: For Stage Binet, the correct values are A 3 B 9 C 2 for responders, and A 2 B 22 C 5 for non-responders after 2nd dose. For CIRS  after the 3rd dose, those with CIRS<3 and not-responders are 13 p-value is 0,97. In the “type of treatment “ line, patients not responders after the 2nd dose treated with BTKi are 15. Patients not on rituximab are 7 responders after the 2nd dose and 10 non-responders p-value 0,19;  after the 3rd dose 2 are responders and 3 non-responders p-value 0,13. [Table1]
  2. We added this figure to represent neutralizing IgG titers [Figure 1]
  1. We provide the following data: “there is no correlation between IgG and IgA levels and anti SARS-CoV2 neutralizing antibodies after II dose: Spearman r 0.01212 95% CI -0.3092 to 0.3310 p= 0,9408 for IgG;  Spearman r 0,2029 95% CI -0.1254 to 0.4912 p= 0,2092 for IgA. While a correlation is found with IgM levels:  Spearman r 0,3275 95% CI 0.008189 to 0.5862 p= 0,0391.” [line 106-109]
  2. In the text there is a typo. The correct number of patients that received the third dose is 39 because two patients died before and one refused. Of those 27 were analyzed, so patients not analyzed are 12, not 13. [145-146]
  3. Of 27 analyzed patients after the third dose 5 were previously responders, and of those one became negative. [line 146-150]
  4. As suggested we provide the following considerations:

 “Randomized controlled trials demonstrated that the BNT162b2 vaccine is highly effective in healthy population with a response rate of 91% after two doses [13]). Patients with hematological malignancies had different seroconversion rate, ranging from 94% in myelodysplastic syndrome and Hodgkin lymphoma to 47% in CLL.  [line 164-167]

“After the 2nd dose, untreated patients had a response rate of 60% versus 23% in treated ones, underling the previously reported difference in seroconversion. Our data globally agree with the findings of Roeker LE et al. in a similar cohort that reported a seroconversion rate of 23% in treated patients (21% BTKi; 14% anti-CD20 MoAb; 0% Ven-R)” [line 173-176].

“This finding agrees with what was reported by Kohn M et al. in 33 CLL patients treated with anti-CD20 MoAb[21]. Moreover, Bagacean et al. reported a global response rate of 35% after administration of the third dose in patients that failed the second with a lower rate (24%) for patients on active treatment[9]. Herishanu et al. reported a rate of 24% that was lower in patients o treatment with BTKi (15%) and Ven-R (7%)[8].[line 187-192]

Minor comments:

  1. We accepted your suggestion and we stressed the requested point by adding a more appropriate citation [line 49-51]
  2. We accepted your suggestion and we added the requested information [line 71-72]
  3. We revised the article in order to improve the language

Round 2

Reviewer 2 Report

The authors have addressed all my comments and have significantly improved their manuscript. There appears to be still one final mistake in the table. The number of non-responder patients after 3rd dose in the parameters number of treatments equals 20 individuals and still does not add up to 22. Also in sentence 175, the word "underling" (which is not a verb) should be "underlining", I think. Please check and update accordingly.

Author Response

Dear Sir/Madam 

I would like to thank you for your prompt review and detailed analysis of the manuscript. Here we provided corrections of the mistakes.

Best regards

Andrea Galitzia